# Post-Harvest Treatment with Methyl Jasmonate Impacts Lipid Metabolism in Tomato Pericarp (*Solanum lycopersicum* L. cv. Grape) at Different Ripening Stages

**DOI:** 10.3390/foods10040877

**Published:** 2021-04-16

**Authors:** Silvia Leticia Rivero Meza, Eric de Castro Tobaruela, Grazieli Benedetti Pascoal, Isabel Louro Massaretto, Eduardo Purgatto

**Affiliations:** 1Department of Food Science and Experimental Nutrition/Food Research Center, Faculty of Pharmaceutical Sciences, University of São Paulo (USP), Av. Prof. Lineu Prestes 580, bl 14, Butantã, 05508-000 São Paulo, SP, Brazil; silviarm@usp.br (S.L.R.M.); erictobaruela@gmail.com (E.d.C.T.); isabelmassaretto@gmail.com (I.L.M.); 2Faculty of Medicine, Federal University of Uberlândia (UFU), Av. Pará, 1720, bl 2U, Umuarama, 38405-320 Uberlândia, MG, Brazil; grazi.nutri13@gmail.com

**Keywords:** post-harvest treatment, jasmonate, metabolite profiling, lipid metabolism, *Solanum lycopersicum*, ethylene inhibition, fruit quality

## Abstract

The application of exogenous jasmonate can stimulate the production of ethylene, carotenoids, and aroma compounds and accelerate fruit ripening. These alterations improve fruit quality and make fruit desirable for human consumption. However, fruit over-ripening results in large losses of fruit crops. This problem is overcome by applying 1-methylcyclopropene to the fruits, due to its capacity to block the ethylene receptors, suppressing fruit ripening. In this study, treatments with only 1-methylcyclopropene and both 1-methylcyclopropene and methyl jasmonate were administered to observe whether exogenous methyl jasmonate can improve the metabolite levels in fruits with blocked ethylene receptors. Fruit pericarps were analyzed at 4, 10, and 21 days after harvest (DAH) and compared with untreated fruits. The post-harvest treatments affected primary metabolites (sugars, organic acids, amino acids, and fatty acids) and secondary metabolites (carotenoids, tocopherols, and phytosterols). However, the lipid metabolism of the tomatoes was most impacted by the exogenous jasmonate. Fatty acids, carotenoids, tocopherols, and phytosterols showed a delay in their production at 4 and 10 DAH. Conversely, at 21 DAH, these non-polar metabolites exhibited an important improvement in their accumulation.

## 1. Introduction

At the onset of tomato ripening, changes in primary metabolites are observed, such as the accumulation of glucose and fructose and the presence of citric and malic acids in ripe fruits [1]. Sugars and organic acids are critical to good flavor, contributing to sweetness and acid balance. Consequently, they are responsible for consumer acceptance [2].

Additionally, changes in secondary metabolites of tomato fruit are observed mainly for those related to health benefits as well as lycopene, β-carotene, α-tocopherol, β-tocopherol, and β-sitosterol. Carotenoids and tocopherols play an important role in human nutrition, mainly due to antioxidant properties and the visual perception of ripe fruits, while phytosterols are associated with reducing LDL cholesterol and total cholesterol [3,4,5]. Many of these ripening processes are regulated by plant hormones such as ethylene, methyl jasmonate, abscisic acid, and other phytohormones [6,7].

Methyl jasmonate can interact with other phytohormones, such as ethylene, in promoting biological activity, such as antibacterial and antifungal activities and signaling plant defenses [8]. Application of exogenous jasmonate stimulates ethylene production, the degradation of chlorophyll, accumulation of β-carotene, and production of aroma compounds, which can accelerate fruit ripening [9].

Although these changes can improve the fruit quality, making it desirable for consumption, fruit over-ripening can result in large losses of fruit crops. This problem can be overcome by exogenous application of 1-methylcyclopropene to tomato fruits, due to its ability to reduce the ethylene production and respiration rate of climacteric fruits [10]. This action prolongs the shelf life of tomato fruits by retaining their firmness and delaying lycopene production and consequently color development [11,12]. This study investigates the metabolic response to methyl jasmonate applied concomitantly with 1-methylcyclopropene to harvested tomato fruits during their ripening.

## 2. Materials and Methods

### 2.1. Plant Material and Post-Harvest Treatment

Tomatoes (*Solanum lycopersicum* cv. Grape) at the mature green stage (*N* = 1200) were collected from a standard commercial greenhouse in Ibiúna (23°39′21″ S; 47°13′22″ W), São Paulo, Brazil. Fruits were sterilized with 0.1% aqueous sodium hypochlorite solution for 15 min. Four biological replicates were applied in the experiment, each comprising 100 fruits. Tomatoes were randomly separated into 3 groups (*N* = 400 by group): (1) control group (CTRL), with no treatment; (2) treated with 1-methylcyclopropene (MCP); (3) treated with both 1-methylcyclopropene and methyl jasmonate (MCP+MeJA). Fruits were left to ripen spontaneously in a 323 L chamber at a constant temperature (20 ± 2 °C) and humidity (80% ± 5% RH) in a 16 hour-day/8-hour-night cycle. For the MCP treatment, the instructions of the manufacturer for “manual addition” were followed: 2.45 g of 1-methylcyclopropene (powder, 3.3% w/w active ingredient, SmartFresh post-harvest treatment; AgroFresh Solutions, Inc., Philadelphia, PA, USA) was weighed and transferred to a 500 mL Erlenmeyer flask capped with a rubber stopper. Using a syringe, 75 mL of de-ionized water was added to the flask, dissolving the powder, and releasing the 1-methylcyclopropene gas. The flask was placed in the chamber, the stopper was removed, and the chamber was closed immediately. A small fan was placed in the chamber, directed at the flask to aid in the dispersion of the gas. For the MCP+ MeJA group, methyl jasmonate (Sigma-Aldrich, Saint Louis, MO, USA) was applied to a filter paper left on the chamber wall for evaporation (100 ppm, final concentration in gas phase), and 1-methylcyclopropene treatment was made as described above. Both treatments were conducted for the second time 12 h after the first exposure to the hormone, totaling 24 h of treatment. Samples of 10 fruits from each replicate were randomly taken at 4, 10, and 21 days after harvest (DAH), considering the control group as a reference. Biological replicates were composed of pericarp tissues by removing the placenta and fruit seeds. The pericarp samples were frozen in liquid nitrogen and stored at −80°C for subsequent analyses.

### 2.2. Ripening Parameters

#### 2.2.1. Ethylene Emission

Ethylene emission was analyzed by placing five tomato fruits in 600 mL airtight glass containers at 25 °C for 1 h. Five 1 mL samples of gas produced in the headspace were then collected with gastight syringes through a rubber septum. A gas chromatograph with a flame ionization detector (Flame Ionization Detector (FID) for GC; (Agilent Technologies, Santa Clara, CA, USA, model HP-6890) and HP-PLOT Q column (30 m × 0.53 mm × 40 µm) were used to evaluate ethylene emission. The injector and detector temperatures were equally established at 250 °C and the oven at 30 °C. The helium gas flow was set at 1 mL.min^−1^, and the injections were performed using a pulsed splitless method.

#### 2.2.2. Fruit Surface Color

Fruit surface color was measured using a HunterLab ColorQuest XE colorimeter (Hunter Associates Laboratory, Inc.) in terms of L*, a*, and b* space. The experimental data were treated to obtain values of the hue angle. Three measurements were made at the equatorial zone of six tomato fruits [13].

### 2.3. Analysis of Metabolite Profiling of Tomato Fruit Using GC-MS

#### 2.3.1. Extraction and Derivatization of Polar Metabolites

The extraction and derivatization of polar metabolites were conducted as described in [14]. For the extraction process, 100 mg of frozen pericarp powder was mixed with 100% distilled methanol at −20 °C (1400 μL) and ribitol (200 μg.mL^−1^, internal standard; 60 μL). The mixture was vortexed, incubated in a thermomixer at 950 rpm for 10 min at 70 °C, centrifuged at 11,000× *g* for 10 min, and the supernatant was collected. To the upper phase was added chloroform at −20 °C (750 μL) and Milli-Q water (1500 μL), followed by mixing and centrifugation at 2200× *g* for 15 min. The upper hydrophilic phase (150 μL) was collected and dried under nitrogen gas. Sample derivatization comprised adding 20 mg.mL^−1^ methoxyamine hydrochloride (Sigma-Aldrich, St. Louis, MO, USA; 40 μL) and pyridine with subsequent incubation in an orbital shaker at 1000 rpm and 37 °C for 2 h. Consecutively, N-methyl-N-(trimethylsilyl) trifluoroacetamide (MSTFA; 70 μL) was added to the sample, followed by incubation in an orbital shaker at 1000 g and 37 °C for 30 min. Finally, the derivatized samples were moved into glass vials and analyzed by GC-MS. A pool of polar metabolite external standards (1 mg.mL^−1^, Sigma-Aldrich) was applied to certify the identified metabolites by mass spectral comparison: D-glucose; D-fructose; maltose; sucrose; D-galactose; myo-inositol; citric acid; L-alanine; L-serine; L-proline; L-aspartate; L-glutamate [15].

#### 2.3.2. Extraction and Derivatization of Non-Polar Metabolites

For the extraction process, 1000 mg of frozen pericarp powder was mixed with chloroform (1250 μL), methanol (2500 μL), and n-tridecane (800 μg.mL^−1^, internal standard; 20 μL), followed by vortexing for 10 s and incubation on ice for 30 min. Then, 1.5% sodium sulfate (1250 μL) and chloroform (1250 μL) were added to the mixture, incubated on ice for 5 min and centrifuged at 4 °C for 1000× *g* and 15 min. The upper polar phase was collected and dried under nitrogen gas. The sample was redissolved in hexane (1000 µL), toluene (200 µL), methanol (1500 µL), and 8% chloridric acid (300 µL), mixed for 10 s, and incubated for 1.5 h at 100 °C. Subsequently, hexane (1000 μL) and Milli-Q water were added to the sample and mixed [16,17,18]. The hexane phase was separated and dried under nitrogen gas. The sample was redissolved in hexane (80 μL) and pyridine (20 μL) and derivatized with MSTFA (40 μL). Finally, the derivatized samples were moved into glass vials and analyzed by GC-MS. A pool of fatty acid methyl ester (FAME) external standards (Sigma-Aldrich) was applied to certify the identified metabolites by mass spectral comparison: methyl laurate (C12:0, 0.8 mg.mL^−1^); methyl tetradecanoate (C14:0, 0.8 mg.mL^−1^); methyl palmitate (C16:0, 0.8 mg.mL^−1^); methyl octadecanoate (C18:0, 0.4 mg.mL^−1^); methyl arachidate (C20:0, 0.4 mg.mL^−1^); methyl docosanoate (C22:0, 0.4 mg.mL^−1^); methyl lignocerate (C24:0, 0.4 mg.mL^−1^); methyl linoleate (C 18:2, 0.4 mg.mL^−1^); (Z)-9-oleyl methyl ester (C 18:1, 0.4 mg.mL^−1^); methyl linolenate (C 18:3, 0.4 mg.mL^−1^); methyl palmitoleate (C 16:1, 0.8 mg.mL^−1^) [15].

#### 2.3.3. GC-MS Analysis

Derivatized samples were analyzed by GC-MS (Agilent 5977 Series GC/MSD, Agilent Technologies, Santa Clara, CA, USA) [15]. Trimethylsilyl derivatives (1 μL) were injected into an injector at 230 °C in splitless mode. The oven temperature ramp applied was 80 °C (initial temperature), held for 2 min, heated at 15 °C.min^−1^ to 330 °C, and held for 6 min. The electron impact ionization mass spectrometer was set at ionization voltage 70 eV; ion source temperature 250 °C; injection port temperature 250 °C; and mass scan range 70–600 m/z at 20 scans.s^−1^. The column used was an HP5ms column (30 m × 0.25 m × 0.25 μm). The flow rate of helium gas was 2 mL.min^−1^. Acquisition, deconvolution, and analyses of experimental data were processed by MassHunter Quantitative Analysis software (Agilent, CA, EUA). The NIST mass spectral library (NIST 2011, Gaithersburg, MD, USA) was used for retention index (RI) comparison and data validation. Some of the identified metabolites were also confirmed by mass spectral comparison with the authentic external standards previously described.

### 2.4. Analysis of Carotenoids by HPLC

Frozen pericarp powder (200 mg) was mixed with 100 µL of 30% NaCl (w:v) solution and 200 µL of dichloromethane to extract carotenoids. Hexane:ether (1:1; 500 µL) was added to the mixture and centrifuged at 13,000× *g* at 4 °C for 5 min. This protocol was repeated thrice, and the organic phases were pooled [19]. The upper phase was dried under nitrogen gas and dissolved in ethyl acetate. Samples were analyzed by HPLC (Analytical HPLC, 1260 Infinity II LC System; Agilent Technologies, Santa Clara, CA, USA), coupled to a diode array detector (DAD), and equipped with a YMC Carotenoid HPLC C30 (5 µm × 250 mm × 4.6 mm) column [20]. Lycopene, β-carotene, and lutein from Sigma-Aldrich were used as external standards.

### 2.5. Statistical Analysis

Experimental data were expressed as mean ± standard deviation (SD) of four biological replicates. Statistical analysis was performed by one-way analysis of variance (ANOVA), and Tukey’s test was applied to establish significant differences among mean values at *p* < 0.05, using the Minitab 19.0 software package (State College, PA, USA). For multivariate analysis, raw data were normalized by the internal standard area, processed using log transformation (log 2), mean-centered, and divided by the square root of the deviation of each variable (Pareto scaling). Principal component analysis (PCA), heatmaps, and fold-change analysis were executed to evaluate differences between treated and untreated groups, using the MetaboAnalyst 4.0 server (https://www.metaboanalyst.ca accessed on 15 April 2021) [21].

## 3. Results and Discussion

### 3.1. Effect of Methyl Jasmonate on the Ethylene Emission and Fruit Surface Color of Tomatoes

In this study, the alterations of metabolites identified in tomato fruits under post-harvest treatments were observed. Therefore, one group of fruits with ethylene inhibited by 1-methylcyclopropene were exposed to methyl jasmonate hormone (MCP+MeJA), other fruit groups were treated only with 1-methylcyclopropene (MCP), and the untreated tomato fruits (CTRL) were used as a reference for the assays. The three groups of fruits are visualized in Figure 1A.

The CTRL-group fruits achieved the breaker stage at 4 DAH and the ripe stage at 10 DAH. Regarding treated fruits, the breaker and red stages were achieved with MCP at 13 and 21 DAH, respectively, and MCP + MeJA at 10 and 13 DAH, respectively. The ripening stages of the CTRL group were characterized by measuring the ethylene emission and surface color of the tomato fruits from the day of harvest to 21 DAH (Figure 1B,C). The metabolite profiling was analyzed at 4, 10, and 21 DAH, aiming to observe the effect of treatments compared to the CTRL.

Treatments with both 1-methylcyclopropene and methyl jasmonate, and only 1-methylcyclopropene, showed a delay in fruit ripening by reducing ethylene emission and fruit surface color, as compared to the CTRL group. A similar result was observed in a study where tomatoes were treated with 1-methylcyclopropene, which reported a reduction in ethylene emission and respiration rate [22]. Both groups, MCP and MCP + MeJA, presented the characteristic curves of ethylene emission of climacteric fruits. Fruits treated only with 1-methylcyclopropene showed the longest delay in fruit ripening, characterized by their ethylene peak and redness at 21 DAH. However, tomatoes treated with both 1-methylcyclopropene and methyl jasmonate showed an ethylene peak at 13 DAH when they acquired a reddish color.

It was observed that using exogenous methyl jasmonate hormone in fruits with ethylene receptors blocked by 1-methylcyclopropene stimulated the ripening process, as compared to fruits treated only with 1-methylcyclopropene. This behavior indicates that 1-methylcyclopropene efficiently blocks ethylene receptors and consequently may avoid the interaction of ethylene with other phytohormones related to ripening processes such as endogenous methyl jasmonate, delaying fruit ripening. However, when exogenous methyl jasmonate hormone was applied to these fruits, an acceleration in ripening was observed by the accumulation of pigments and anticipation of an ethylene peak from 21 to 13 DAH. Additionally, the highest peak of ethylene emission was observed for the MCP + MeJA group, which may be related to stimulation of ethylene biosynthesis in climacteric fruits by methyl jasmonate hormone. Thus, our results suggest that exogenous methyl jasmonate can act independently of ethylene, or the blockage of ethylene receptors was reversed after some time. Therefore, the synthesis of new receptors in tomato fruits after 1-methylcyclopropene treatment could be possible, as this occurs in several fruits [22,23]. This behavior may be responsible for the increased ethylene production after some time, as observed after 10 DAH.

### 3.2. Primary Metabolite Profiling Affected by Post-Harvest Hormone Treatment

Primary metabolites are important components related to fruit quality. Additionally, they are considered crucial for plant growth and development. Thus, understanding the fruit metabolism can support developing future approaches for its manipulation [24]. In this work, a total of 46 primary metabolites were identified by GC-MS metabolomics analysis: 10 sugars (glucose, fructose, sucrose, allose, gulose, glucaric acid, myo-inositol, mannose, ribose, and arabinofuranose); 9 organic acids (oxaloacetic, citric, succinic, aconitic, malic, citraconic, fumaric, propanoic, and butanoic acids); 12 amino acids (proline, serine, valine, threonine, aspartic acid, glutamic acid, glutamine, γ-aminobutyric acid (GABA), asparagine, tryptophan, phenylalanine, and tyrosine); 12 saturated fatty acids (capric, lauric, myristic, palmitic, stearic, eicosanoic, docosanoic, tricosanoic, lignoceric, hyenic, cerotic, and montanic acids); 3 unsaturated fatty acids (oleic, linoleic, and linolenic acids) at 4, 10, and 21 DAH (Table 1). Table 1 shows the effects of methyl jasmonate and 1-methylcyclopropene on the accumulation or reduction of each metabolite at the three different maturation stages, indicated by the area normalized by the internal standard.

Moreover, a global overview of the metabolic changes occurring in tomatoes during ripening was obtained to evaluate significant differences among accumulated metabolites in treated fruits compared with the control group (Figure 2).

A PCA was performed on primary metabolites at the 4th, 10th, and 21st ripening stages, confirming the high reproducibility among the four biological replicates and groups analyzed. Moreover, clear separation of the CTRL group and both treated groups was evidenced for the primary metabolites in the PCA score. Heatmap analysis was used to analyze the differences between treated and untreated groups regarding the metabolite changes on each day after harvest (Figure 3, Figure 4 and Figure 5).

Primary metabolism is essential for fruit quality. Sugars, organic acids, and amino acids are responsible for the taste of tomato fruits, facilitating sensory perception. Amino acids and fatty acids play important roles as precursors of aroma compounds [7]. Treatment with 1-methylcyclopropene impacted sugar and organic acids, inhibiting their production during ripening. Fruits treated only with 1-methylcyclopropene were most affected, showing a greater delay in accumulating sugars and organic acids than fruits treated with both 1-methylcyclopropene and methyl jasmonate (Figure 3). For instance, glucose showed a 22-, 13-, and 23-fold reduction at 4-, 10-, and 21 DAH, respectively, in MCP, as compared to the CTRL. Mannose, ribose, and malic and aconitic acids exhibited a 14-, 30-, 21-, and 20-fold decrease in levels at 4 DAH, respectively. Conversely, fructose, sucrose, and citraconic acid showed a 12-, 15-, and 27-fold decrease in levels at 10 DAH when MCP was compared with the CTRL (Table 1). Reductions in the levels of these metabolites in fruits treated with 1-methylcyclopropene are shown in (Figure 2), based on the fold-change analysis of the treated fruits and control group.

Exceptionally, glucose, glucaric acid, and mannose levels showed an increase at 10 DAH in MCP + MeJA, as compared to the CTRL. Similar behavior was observed for myo-inositol, propanoic, and butanoic acids at 21 DAH (Table 1). As shown in Figure 3, heatmap analysis demonstrated a tendency of these metabolites to increase at 10 DAH. As observed by ethylene emission, the minor impact on the production of sugars and organic acids observed for MCP + MeJA may suggest that methyl jasmonate plays an important role in ripening. This may act independently of endogenous ethylene, stimulate the synthesis of new receptors, or reverse the blockage of ethylene receptors after some time.

Amino acid profiling was also affected by the action of 1-methylcyclopropene. Inhibition of the production of amino acids during ripening was observed for both MPC and MCP + MeJA compared with the control (Figure 4). The most affected amino acids were aspartic acid at 4 DAH and GABA at 10 DAH, showing a 28- and 10-fold reduction in their levels with MCP, respectively. However, MCP + MeJA showed 11- and 14-fold decreases, respectively, as compared to the CTRL, as shown in (Figure 2). Conversely, tyrosine and phenylalanine showed levels two- and ninefold higher for MCP and MCP + MeJA at 4 DAH, as compared to the CTRL (Table 1, Figure 2). It is important to highlight that phenylalanine and tyrosine are aromatic amino acids, which participate in the shikimate pathway and are responsible for the aroma development of fruit. Table 1 shows that the total amino acid level was represented mostly by proline, glutamic, and aspartic acids, which are important to fruit quality, as they provide sweetness and umami flavor.

Additionally, fatty acid profiling was also affected by the post-harvest treatments. The action of 1-methylcyclopropene showed a greater impact on fatty acids such as oleic, capric, lauric, palmitic, stearic, and myristic acids at 10 DAH, as shown in (Figure 5), decreasing their levels by 17-, 10-, 14-, 17-, 14-, and 14-fold in the MPC group, respectively, and 7-, 6-, 9-, 11-, 1-, and 7-fold in the MCP + MeJA group, respectively, as compared to the CTRL (Table 1). The reduction in fatty acids by 1-methylcyclopropene was evident when the fold-change analysis was applicable (Figure 2). The MCP + MeJA group also showed a reduction in fatty acid levels; however, this was less impactful than the MCP group (Figure 5). The most impacted were the linoleic and myristic acids at 4 DAH with a reduction of 119- and 26-fold in MCP, respectively, and a 23- and 9-fold decrease in MCP + MeJA, respectively, as compared to the CTRL (Table 1).

Conversely, a tendency of increased levels of some fatty acids was also detected, as well as in lignoceric, cerotic, and α-linolenic acids at 4 DAH, and palmitic and linoleic acids at 21 DAH for the MCP and MCP + MeJA groups (Figure 5). In the MCP group, an increase was detected in the levels of lignoceric and α-linolenic acids at 4 DAH by 7- and 4-fold, respectively, while in MCP + MeJA, the increases were 28 and 3-fold, respectively. Moreover, palmitic and linoleic acids were increased by 2- and 10-fold, respectively, in MCP, and 3- and 10-fold, respectively, in MCP + MeJA at 21 DAH (Table 1). Interestingly, the MCP + MeJA group was less impacted when reductions were observed and more impacted when increases were observed compared with the MCP group. This behavior may indicate that methyl jasmonate can act as a stimulator in fatty acid production. Palmitic and eicosanoic acids contributed crucially to the total saturated fatty acid level, and oleic and linoleic acids contributed to the total unsaturated fatty acid level, which is notable, as they play an important role in the fruit quality and nutritional value.

### 3.3. Secondary Metabolite Profiling Affected by Post-Harvest Hormone Treatment

The secondary metabolites identified in tomato fruits at 4, 10, and 21 DAH were lycopene, β-carotene, and lutein by HPLC analysis; α-tocopherol, β-tocopherol, γ-tocopherol, phytol, β-sitosterol, stigmasterol, and stigmastadienol were identified by GC-MS analysis.

Lycopene was the most affected by the action of 1-methylcyclopropene, reducing its level not only in MCP but also in MCP + MeJA by 29- and 25-fold, respectively, at 4 DAH. However, at 10 DAH, lycopene was reduced by eight- and sixfold, respectively, compared with the CTRL (Figure 6A); β-carotene and lutein showed a decrease less than threefold by 1-methylcyclopropene at the ripening stages (Appendix A). These remarkable impacts on the synthesis of carotenoids are illustrated in (Figure 2), mainly at the onset of ripening.

However, the action of 1-methylcyclopropene had a lesser impact on lycopene accumulation at 21 days of hormone treatment, decreasing its production by 2.8-fold, and its action was completely reversed by the methyl jasmonate hormone. Fruits treated with methyl jasmonate showed an increase not only in lycopene production but also in β-carotene and lutein at 21 DAH, indicating the important role that methyl jasmonate plays in the synthesis of carotenoids (Figure 6A). Lycopene and β-carotene showed an increase of 10%, and lutein of 20%, as compared to the CTRL (Figure 6A, Appendix A), which is considered relevant since these bioactive compounds have been associated with health benefits, leading to decreases in the occurrence of chronic non-communicable diseases [25]. The total carotenoid level was represented mainly by lycopene.

Tocopherol profiling showed similar behavior to carotenoids during ripening, decreasing its levels in both treated groups at 4 and 10 DAH (Figure 2). At 21 DAH, it presented a decrease, in the MCP group, and an increase, in MCP+MeJA, of tocopherols, as compared to the CTRL. The α-tocopherol levels showed a reduction in MCP and MCP + MeJA of 5- and 4-fold, respectively, at 4 DAH, while at 10 DAH, they decreased by 12 and 3-fold, respectively. The β-tocopherol levels showed a reduction of 14 and 12-fold at 4 DAH, and 23- and 9-fold at 10 DAH in MCP and MCP+MeJA, respectively. Additionally, γ-tocopherol was decreased by 6-fold at 4 and 10 DAH in both treatment groups, except for the MCP + MeJA at 10 DAH, which decreased 1.7 fold, as compared to the CTRL (Figure 6B, Appendix A).

Conversely, at 21 DAH, tocopherol profiling was less affected by 1-methylcyclopropene and positively impacted by the concomitant treatment of 1-methylcyclopropene and methyl jasmonate, showing increases of 40% in the α-tocopherol and β-tocopherol levels and 21% in the γ-tocopherol levels, as compared to the CTRL (Figure 6B, Appendix A). The total tocopherol level was characterized mainly by the α-tocopherol content. An acyclic diterpenoid identified was phytol, which presented a twofold reduction in MCP at 4 DAH and a twofold increase in MCP + MeJA at 10 DAH (Figure 6B, Appendix A). The impact of these treatments at 4 and 10 DAH is also shown in (Figure 2).

Phytosterols were also affected by 1-methylcyclopropene, showing fivefold reductions in β-sitosterol levels in MCP at 4 and 10 DAH and threefold reductions in MCP + MeJA at 4 and 10 DAH, as compared to the CTRL. Stigmasterol exhibited four and sevenfold reductions in MCP at 4 and 10 DAH, respectively, while MCP + MeJA showed three and fivefold decreases at 4 and 10 DAH, respectively. Stigmastadienol was the phytosterol most affected by 1- methylcyclopropene, decreasing ninefold at 4 DAH (Figure 6C, Appendix A); β-sitosterol and stigmasterol were the major sources of the total phytosterol level. Moreover, down-regulation exceeding twofold, as compared to the CTRL, is observed for the phytosterols in (Figure 2). Divergent behavior of phytosterols profiled at 4 and 10 DAH, β-sitosterol, stigmasterol, and stigmastadienol showed an increase in their levels by 42%, 34%, and 32%, respectively, in fruits treated with both 1-methylcyclopropene and methyl jasmonate at 21 DAH (Figure 6C, Appendix A).

### 3.4. Lipid Metabolism Affected by the Post-Harvest Jasmonate Treatment

The metabolite profiling of the tomato fruit pericarp treated only with 1-methylcyclopropene and with both 1-methylcyclopropene and methyl jasmonate showed a significant impact on the fruit quality and, consequently, the ripening process. Although the profiles of sugars, organic acids, and amino acids were affected by the jasmonate treatment, the most remarkable difference observed was in the lipid metabolism.

Fruits treated with methyl jasmonate showed a positive impact on the accumulation of metabolites, mainly in the non-polar metabolites (fatty acids, carotenoids, tocopherols, and phytosterols). The application of exogenous methyl jasmonate in fruits with blocked ethylene receptors showed that methyl jasmonate could act independently of endogenous ethylene, suggesting that the blocking of ethylene receptors was reversed after 10 DAH, or new ethylene receptors were synthesized. Post-harvest treatment with jasmonate showed that it is possible to obtain an improved fruit quality with a prolonged shelf life.

Oleic, capric, lauric, palmitic, stearic, and myristic acids showed a 17-fold reduction when treated with only 1-methylcyclopropene and up to an 11-fold reduction when treated with both methyl jasmonate and 1-methylcyclopropene, as compared to the untreated fruits at 10 DAH. It is noteworthy that a drastic decrease was observed in the levels of linoleic and myristic acids at 4 DAH with both treatments (Table 1, Figure 2). Additionally, reductions in the levels of carotenoids, tocopherols, and phytosterols were also detected at the onset of ripening (Figure 6, Appendix A).

Conversely, an interesting increase in the levels of lignoceric, cerotic, and α-linolenic acids at 4 DAH and palmitic and linoleic acids at 21 DAH with both treatments was detected (Figure 2 and Figure 5). Moreover, notable accumulations in the levels of secondary metabolites, such as lycopene, β-carotene, lutein, α-, β-, and γ-tocopherols, β-sitosterol, stigmasterol, and stigmastadienol, were detected at 21 DAH by the action of methyl jasmonate (Figure 6, Appendix A). However, it is noteworthy that the maturation stage and hormonal regulation may not be the only factors responsible for the improved lipid metabolism in tomato fruits; other factors such as genetic factors, cultural practices, cultivation, and environmental conditions should be considered [25]. Understanding the interactions between hormone treatment and environmental factors, genotype, and agronomic practices is essential to produce high-quality fruits by improving the synthesis of high-value nutrients.

The treatment with methyl jasmonate can induce significant changes in the metabolite profile of tomato fruits during ripening, positively impacting the nutritional and sensory fruit quality. This treatment, associated with the blocking of ethylene receptors with 1-methylcyclopropene, proved effective in avoiding potential effects on the post-harvest life of the fruits due to the increase in ethylene synthesis caused by methyl jasmonate. Our experimental design involved four replicates, and the consistency of the results indicates that the effects have good reproducibility. However, they were conducted only on the Grape cultivar, and it would be interesting to reproduce these treatments in other cultivars to assess the influence of genotype on responses to treatment with methyl jasmonate and 1-methylcyclopropene. From the viewpoint of applicability, the presented protocol has good commercial potential, since the concentrations of methyl jasmonate and 1-methylcyclopropene used were low and, consequently, did not require large volumes of the compounds. The volatility of the compounds and simplicity of the method of exposure of the fruits make the treatments feasible for larger environments, such as commercial chambers. Although both substances are considered generally recognized as safe (GRAS), further studies about the impact on sensory quality would be important to assess consumer acceptability for the treated fruit.

## Figures and Tables

**Figure 1 foods-10-00877-f001:**
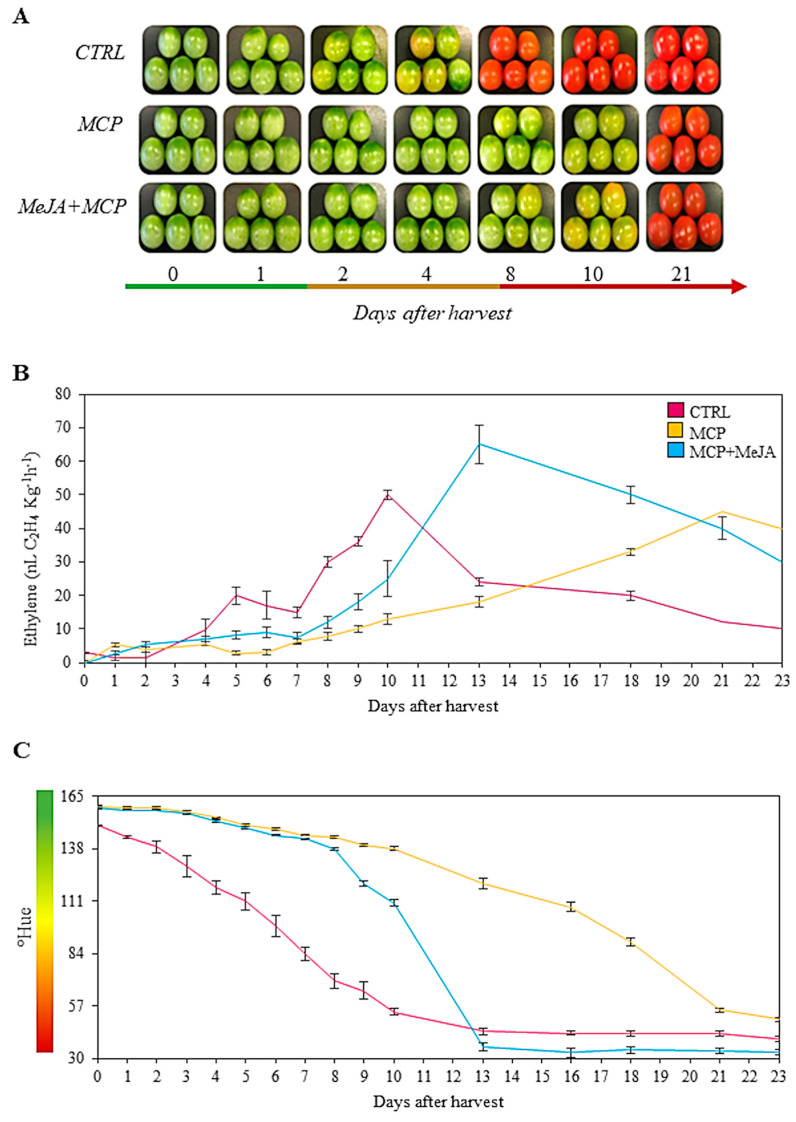
Characterization of tomato (*Solanum lycopersicum* L. cv. Grape) fruits treated with 1-methylcyclopropene (MCP) and both hormones, namely 1-methylcyclopropene and methyl jasmonate (MCP+MeJA), during ripening. (**A**) Representative images of tomatoes. Effects of MCP and MCP + MeJA on ethylene emission (**B**) and fruit color (**C**) compared to the control group (CTRL). Values are means ± standard error of four biological replicates of at least 10 fruits each.

**Figure 2 foods-10-00877-f002:**
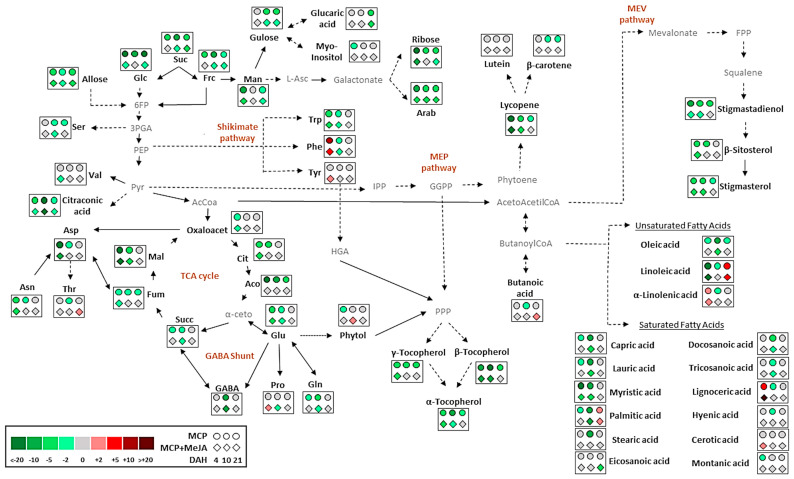
Global overview of metabolic changes occurring in tomato pericarp (*Solanum lycopersicum* L. cv. Grape) treated with 1-methylcyclopropene (MCP) and both hormones, 1-methylcyclopropene and methyl jasmonate (MCP + MeJA), compared to the control group (CTRL). Data were normalized to the CTRL. Metabolites showing up- or down-regulation in each treatment exceeding twofold compared to the CTRL are shown. The color scale displays the different amounts of metabolite in terms of proportional change relative to the level in the appropriate control. Suc, sucrose; Glc, glucose; Frc, fructose; Man, mannose; L-Asc, L-ascorbic acid; Arab, arabinofuranose; 6FP, fructose-6-phosphate; 3-GPA, glyceraldehyde-3-phosphate; Ser, serine; PEP, phosphoenolpyruvate; Trp, tryptophan; Phe, phenylalanine; Tyr, tyrosine; HGA, homogentisic acid; Pyr, pyruvic acid; IPP, isopentenyl pyrophosphate; GGPP, geranylgeranyl; PPP, phytyl pyrophosphate; Val, valine; AcCoA, acetyl-CoA; Oxaloacet, oxaloacetic acid; Cit, citric acid; Aco, aconitic acid; α-keto, α-etoglutaric acid; Succ, succinic acid; Fum, fumaric acid; Mal, malic acid; Glu, glutamic acid; GABA, γ-aminobutyric acid; Gln, glutamine; Pro, proline; Arg, arginine; Asp, aspartic acid; Thr, threonine; Asn, asparagine; FPP, farnesyl pyrophosphate.

**Figure 3 foods-10-00877-f003:**
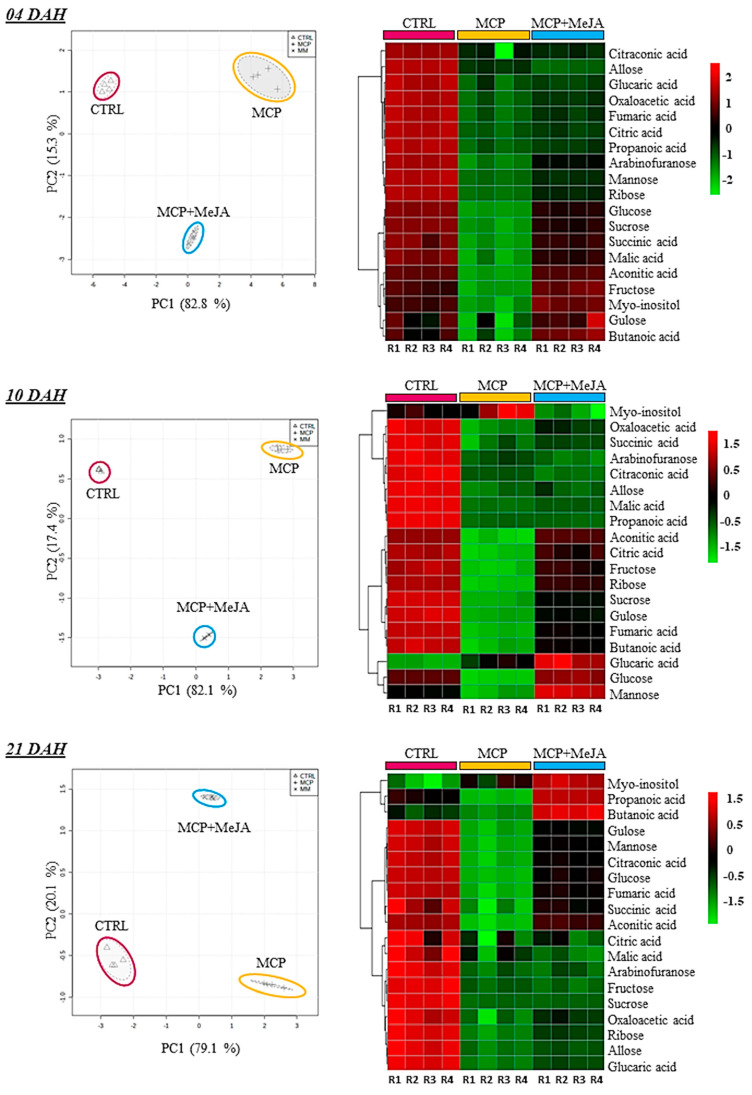
Relative contents of sugars and organic acids in tomato pericarp (*Solanum lycopersicum* L. cv. Grape) exposed to 1-methylcyclopropene (MCP) and both 1-methylcyclopropene and methyl jasmonate (MCP+MeJA) compared to the control group (CTRL). Unsupervised principal component analysis (PCA-score) and heatmap analysis represent the major sources of variability. Color scale represents the variation in the relative concentration of compounds, from low (green) to high (red) contents at 4, 10, and 21 days after harvest (DAH).

**Figure 4 foods-10-00877-f004:**
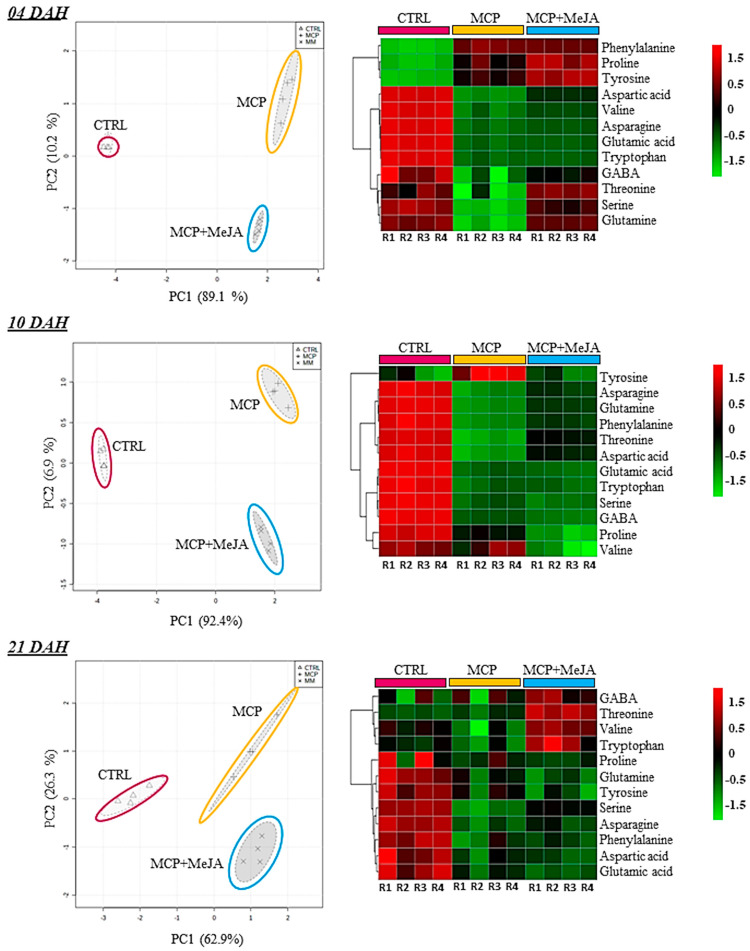
Relative contents of amino acids in tomato pericarp (*Solanum lycopersicum* L. cv. Grape) exposed to 1-methylcyclopropene (MCP) and both hormones, 1-methylcyclopropene and methyl jasmonate (MCP+MeJA), as compared to the control group (CTRL). Unsupervised principal component analysis (PCA-score) and heatmap analysis represent the major sources of variability. Color scale represents the variation in the relative concentration of compounds, from low (green) to high (red) contents at 4, 10, and 21 days after harvest (DAH).

**Figure 5 foods-10-00877-f005:**
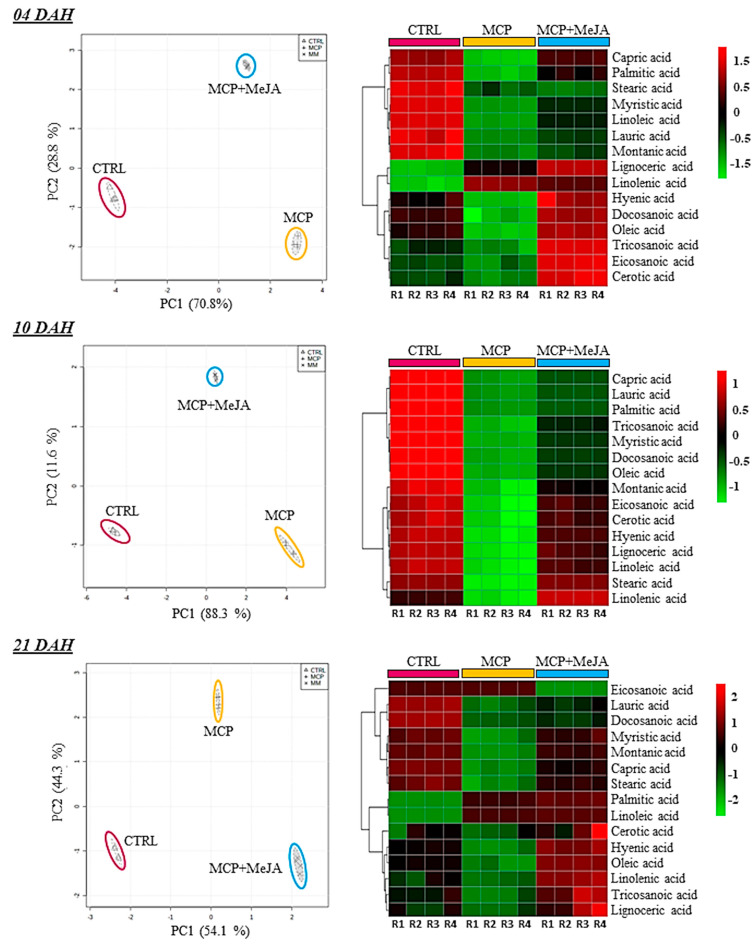
Relative contents of fatty acids in tomato pericarp (*Solanum lycopersicum* L. cv. Grape) exposed to 1-methylcyclopropene (MCP) and both hormones, 1-methylcyclopropene and methyl jasmonate (MCP+MeJA), as compared to the control group (CTRL). Unsupervised principal component analysis (PCA-score) and heatmap analysis represent the major sources of variability. Color scale represents the variation in the relative concentration of compounds, from low (green) to high (red) contents at 4, 10, and 21 days after harvest (DAH).

**Figure 6 foods-10-00877-f006:**
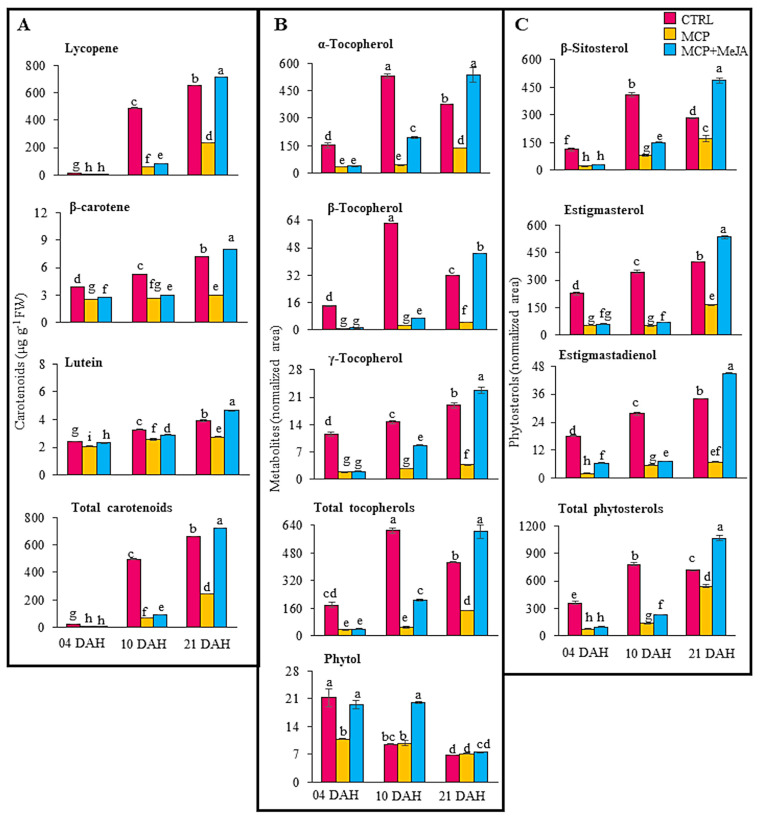
Secondary metabolites in tomato pericarp (*Solanum lycopersicum* L. cv. Grape) exposed to 1-methylcyclopropene (MCP) and both hormones, 1-methylcyclopropene and methyl jasmonate (MCP+MeJA), compared to the control group (CTRL) at 4, 10, and 21 days after harvest (DAH). Contents of carotenoids (**A**), normalized area of tocopherols and phytol (**B**), and phytosterols (**C**). Values are means ± SE of four biological replicates of 10 fruits each. Different letters indicate statistically significant differences (*p* < 0.05).

**Table 1 foods-10-00877-t001:** Primary metabolites in tomato pericarp (*Solanum lycopersicum* L. cv. Grape) exposed to 1-methylcyclopropene (MCP) and both 1-methylcyclopropene and methyl jasmonate (MCP+MeJA) treatments at 4, 10, and 21 days after harvest (DAH), detected by gas chromatography-mass spectrometry (GC-MS).

Metabolite	4 DAH			10 DAH			21 DAH		
	CTRL	MCP	MCP + MeJA	CTRL	MCP	MCP + MeJA	CTRL	MCP	MCP + MeJA
Sugars									
Glucose	1534.5 ± 76.0 ^d^	70.6 ± 4.7 ^g^	598.7 ± 37.9^f^	1977.6 ± 11.4 ^c^	156.5 ± 7.5 ^g^	2977.0 ± 253.0 ^b^	4352.0 ± 281.0 ^a^	189.4 ± 15.2 ^g^	1112.6 ± 140.1 ^e^
Fructose	27474.0 ± 4039.0 ^d^	3935.0 ± 489.0 ^e^	37418.0 ± 5231.0 ^c^	59266.0 ± 6310.0 ^b^	4858.0 ± 544.0 ^e^	26343.0 ± 3352.0 ^d^	101194.0 ± 5662.0 ^a^	25944.0 ± 1592.0 ^d^	24324.0 ± 1808.0 ^d^
Sucrose	38205.0 ± 569.0 ^c^	5105.0 ± 559.0 ^f^	20507.0 ± 1161.0 ^d^	54654.0 ± 716.0 ^b^	3661.0 ± 356.0 ^f^	11961.0 ± 469.0 ^e^	84839.0 ± 4545.0 ^a^	10885.0 ± 358.0 ^e^	11248.0 ± 284.0 ^e^
Allose	1098.6 ± 44.2 ^c^	172.6 ± 12.6 ^ef^	115.1 ± 4.8 ^f^	1563.4 ± 21.5 ^b^	630.6 ± 39.0 ^d^	687.0 ± 56.9 ^d^	3309.0 ± 380.0 ^a^	469.1 ± 49.6 ^d,e^	584.1 ± 27.8 ^d^
Gulose	221.5 ± 9.8 ^d^	196.9 ± 15.1 ^d^	229.8± 10.9 ^d^	790.4 ± 38.3 ^b^	92.6 ± 4.5 ^e^	228.8 ± 10.4 ^d^	1017.0 ± 60.4 ^a^	183.5 ± 17.6 ^d^	404.0 ± 12.9 ^c^
Glucaric acid	42.2 ± 1.6 ^d^	21.9 ± 2.1 ^e^	23.1 ± 0.4 ^e^	72.0 ± 1.30 ^c^	93.4 ± 6.3 ^b^	126.7 ± 8.5 ^a^	124.6 ± 8.7 ^a^	15.1 ± 1.3 ^e^	21.2 ± 0.8 ^e^
Myo-inositol	77.9 ± 2.9 ^e^	33.1 ± 3.2 ^f^	91.3 ± 4.5 ^e^	169.6 ± 2.0 ^d^	178.5 ± 8.1 ^d^	156.6 ± 4.1 ^d^	340.1 ± 20.6 ^c^	412.4 ± 23.9 ^b^	481.0 ± 9.7 ^a^
Mannose	42.2 ± 3.1 ^d^	3.4 ± 0.2 ^f^	6.0 ± 0.3 ^f^	100.0 ± 2.3 ^b^	69.5 ± 3.5 ^c^	148.3 ± 4.7 ^a^	142.4 ± 11.0 ^a^	28.9 ± 2.4 ^e^	66.9 ± 9.7 ^c^
Ribose	174.6 ± 7.4 ^c^	5.9 ± 0.4 ^g^	14.3 ± 0.6 ^g^	249.4 ± 3.9 ^b^	26.2 ± 1.3 ^g^	128.5 ± 3.7 ^d^	386.4 ± 27.1 ^a^	60.7 ± 6.4 ^f^	89.8 ± 3.5 ^f^
Arabinofuranose	15.1 ± 0.7 ^c^	2.1 ± 0.3 ^e^	4.7 ± 0.1 ^e^	25.5 ± 0.8 ^b^	9.3 ± 0.6 ^d^	8.1 ± 0.5 ^d^	45.2 ± 3.3 ^a^	17.2 ± 1.4 ^c^	17.6 ± 1.6 ^c^
Total	68885.0 ± 4082.0 ^c^	9546.0± 771.0 ^e^	59009.0 ± 5831.0 ^c^	118868.0 ± 5993.0 ^b^	9775.0 ± 904.0 ^e^	42764.0 ± 3869.0 ^d^	195750.0 ± 9973.0 ^a^	38206.0 ± 1874.0 ^d^	38348.0 ± 2196.0 ^d^
Organic acids									
Oxaloacetic acid	573.3 ± 24.3 ^e^	173.3± 18.6 ^f^	214.5 ± 6.2 ^f^	2380.4 ± 56.5 ^a^	1234.0 ± 59.1 ^c^	1482.7 ± 55.4 ^b^	1241.9 ± 80.7 ^c^	703.7 ± 72.1 ^d^	823.0 ± 24.2 ^d^
Citric acid	6517.0 ± 413.0 ^c^	765.4 ± 85.1 ^e^	998.9 ± 62.0 ^e^	7878.0 ± 457.0 ^c^	889.9 ± 79.5 ^e^	4117.0 ± 591.0 ^d^	18901.0 ± 1208.0 ^a^	15494.0 ± 1393.0 ^b^	15948.0 ± 747.0 ^b^
Succinic acid	2646.0 ± 360.0 ^c,d^	642.8 ± 49.6 ^f^	1909.3 ± 81.4 ^d,e^	12894.0 ± 485.0 ^a^	2826.0 ± 650.0 ^c^	3587.5 ± 102.0 ^b^	2862.0 ± 254.0 ^b,c^	1851.1 ± 135.1 ^e^	2372.9 ± 71.7 ^c,d,e^
Aconitic acid	61.1 ± 2.9 ^c^	2.9 ± 0.3 ^f^	52.9 ± 2.6 ^d^	83.1 ± 0.9 ^b^	4.2 ± 0.6 ^f^	51.0 ± 1.1 ^d^	101.4 ± 5.70 ^a^	18.2 ± 1.6 ^e^	67.1 ± 2.8 ^c^
Malic acid	2537.5 ± 101.3 ^d^	118.7 ± 5.3 ^f^	156.1 ± 6.2 ^f^	6653.7 ± 174.0 ^c^	1208.0 ± 109.4 ^e,f^	2076.1 ± 46.3 ^d,e^	16800.0 ± 1014.0 ^a^	13024.0 ± 1137.0 ^b^	12285.0 ± 385.0 ^b^
Citraconic acid	17.3 ± 0.7 ^c^	4.1 ± 0.3 ^d^	3.7 ± 0.2 ^d^	104.5 ± 12.4 ^a^	3.9 ± 0.2 ^d^	2.7 ± 0.3 ^d^	101.9 ± 6.1 ^a^	20.9 ± 1.9 ^c^	49.0 ± 2.2 ^b^
Fumaric acid	167.9 ± 6.7 ^c^	61.0 ± 5.3 ^f,g^	73.4 ± 2.7 ^f^	181.7 ± 1.2 ^b^	47.7 ± 2.0 ^h^	99.0 ± 3.0 ^e^	237.8 ± 10.3 ^a^	55.8 ± 5.2 ^g,h^	128.6 ± 5.4 ^d^
Propanoic acid	111.7 ± 6.8 ^c^	14.6 ± 2.1 ^e^	19.0 ± 1.2 ^d,e^	145.6 ± 1.5 ^c^	15.4 ± 0.8 ^e^	15.4 ± 0.5 ^e^	458.6 ± 79.1 ^b^	90.7 ± 5.0 ^c,d^	1318.0 ± 51.3 ^a^
Butanoic acid	284.5 ± 18.9 ^c,d^	221.3 ± 19.7 ^d^	314.9 ± 8.6 ^c,d^	393.4 ± 10.9 ^c^	179.5 ± 8.6 ^d^	274.4 ± 5.9 ^c,d^	641.8 ± 96.9 ^b^	412.7 ± 22.5 ^c^	2404.7 ± 175.3 ^a^
Total	12917.0 ± 804.0 ^d^	2003.4 ± 173.2 ^f^	3743.0±130.7 ^e,f^	30715.0±907.0 ^c^	6409.0±787.0 ^e^	11706.0±691.0 ^d^	41346.0±2535.0 ^a^	31671.0±2748.0 ^c^	35397.0±1239.0 ^b^
Amino acids									
Proline	501.2 ± 14.8 ^d^	798.4 ± 71.8 ^b,c,d^	992.4 ± 55.3 ^b,c^	1167.2 ± 26.8 ^b^	693.5 ± 40.9 ^c,d^	470.9 ± 38.8 ^d^	2863.8 ± 483.0 ^a^	2546.8 ± 166.2 ^a^	2452.7 ± 44.8 ^a^
Serine	78.0 ± 5.7 ^b^	24.8 ± 1.4 ^e^	58.5 ± 4.3 ^c^	93.6 ± 2.3 ^a^	27.7 ± 1.9 ^e^	24.8 ± 1.0 ^e^	49.0 ± 1.5 ^d^	16.1 ± 1.9 ^f^	26.9 ± 1.3 ^e^
Valine	6.2 ± 0.2 ^e^	1.9 ± 0.2 ^f^	2.2 ± 0.1 ^f^	17.6 ± 0.2 ^a^	17.0 ± 0.8 ^a^	14.9 ± 0.5 ^b^	7.9 ± 0.4 ^d^	6.6 ± 0.9 ^e^	9.1 ± 0.3 ^c^
Threonine	6.0 ± 0.45 ^e^	4.3 ± 0.6 ^e^	6.4 ± 0.1 ^d,e^	25.9 ± 0.4 ^b^	9.9 ± 0.5 ^d^	14.9 ± 0.5 ^c^	25.2 ± 0.8 ^b^	26.4 ± 2.7 ^b^	50.5 ± 3.5 ^a^
Aspartic acid	1523.2 ± 54.2 ^b^	54.7 ± 4.8 ^e^	141.7 ± 3.9 ^e^	2166.8 ± 58.7 ^a^	823.2 ± 42.5 ^d^	1163.1 ± 43.0 ^c^	1540.8 ± 161.0 ^b^	1148.6 ± 97.4 ^c^	1116.9 ± 43.0 ^c^
Glutamic acid	1744.3 ± 75.6 ^d^	215.3 ± 17.3 ^e^	228.7 ± 8.3 ^e^	4906.7 ± 42.6 ^a^	1614.2 ± 71.8 ^d^	1530.1 ± 50.3 ^d^	3957.0 ± 424.0 ^b^	2729.0 ± 208.0 ^c^	2551.6 ± 68.0 ^c^
Glutamine	185.1 ± 7.7 ^c^	77.8 ± 5.6 ^e^	172.9 ± 4.8 ^c^	519.3 ± 9.2 ^a^	99.9 ± 5.3 ^d,e^	148.3 ± 4.7 ^c,d^	475.8 ± 38.7 ^a^	351.8 ± 38.2 ^b^	314.9 ± 24.4 ^b^
GABA	207.6 ± 17.2 ^b,c^	148.4 ± 13.2 ^d,e^	173.9 ± 4.5 ^c,d^	1060.8 ± 45.9 ^a^	104.3 ± 4.6 ^e,f^	76.9 ± 4.4 ^f^	216.9 ± 20.6 ^b,c^	221.9 ± 23.6 ^b,c^	242.5 ± 12.0 ^b^
Asparagine	139.7 ± 6.2 ^c^	19.8 ± 1.7 ^e^	22.6 ± 0.5 ^e^	260.2 ± 6.9 ^a^	117.6 ± 5.6 ^d^	142.8 ± 4.8 ^c^	235.6 ± 11.7 ^b^	134.2 ± 10.9 ^c,d^	148.5 ± 11.1 ^c^
Tryptophan	173.7 ± 7.1 ^b^	21.9 ± 1.9 ^d^	22.8 ± 0.9 ^d^	321.4 ± 8.6 ^a^	154.6 ± 7.9 ^c^	154.2 ± 4.9 ^c^	27.1 ± 0.9 ^d^	25.0 ± 1.8 ^d^	31.9 ± 2.8 ^d^
Phenylalanine	7.6 ± 0.5 ^e^	76.2 ± 9.8 ^a^	69.1 ± 2.1 ^a^	26.4 ± 1.3 ^d^	5.9 ± 0.3 ^e^	8.3 ± 0.4 ^e^	50.9 ± 2.7 ^b^	36.3 ± 4.9 ^c^	36.4 ± 1.6 ^c^
Tyrosine	21.1 ± 0.8 ^e^	47.8 ± 3.6 ^c^	71.0 ± 3.9 ^a^	27.8 ± 2.3 ^d,e^	35.7 ± 2.0 ^d^	27.7 ± 1.1 ^d,e^	62.3 ± 3.7 ^b^	51.9 ± 5.0 ^c^	46.4 ± 4.3 ^c^
Total	4593.8 ± 170.8 ^d^	1491.4 ± 125.1 ^f^	1962.3 ± 70.7 ^f^	10593.8 ± 94.5 ^a^	3703.5 ± 149.7 ^e^	3777.0 ± 148.3 ^d,e^	9512.0 ± 940.0 ^b^	7294.0 ± 482.0 ^c^	35397.0 ± 1239.0 ^b^
Saturated fatty acids									
Capric acid	61.9 ± 1.9 ^a^	26.6 ± 0.6 ^c^	52.2 ± 0.7 ^b^	15.0 ± 0.4 ^d^	1.5 ± 0.1 ^g^	2.3 ± 0.0 ^g^	6.8 ± 0.0 ^e^	4.0 ± 0.1 ^f^	5.7 ± 0.1 ^e,f^
Lauric acid	40.5 ± 2.1 ^a^	18.9 ± 0.1 ^c^	22.4 ± 0.3 ^b^	19.7 ± 1.2 ^c^	1.4 ± 0.1 ^e^	2.1 ± 0.0 ^e^	6.8 ± 0.0 ^d^	5.6 ± 0.1 ^d^	5.9 ± 0.1 ^d^
Myristic acid	19.0 ± 0.8 ^a^	0.8 ± 0.0 ^f^	2.1 ± 0.0 ^e^	15.0 ± 0.5 ^b^	1.3 ± 0.1 ^f^	2.6 ± 0.0 ^e^	6.1 ± 0.1 ^c^	4.9 ± 0.1 ^d^	5.9 ± 0.1 ^c^
Palmitic acid	436.7 ± 10.4 ^a^	148.7 ± 5.3 ^e^	285.9 ± 23.0 ^d^	413.1 ± 10.9 ^a,b^	23.9 ± 0.9 ^f^	36.2 ± 0.3 ^f^	159.6 ± 1.4 ^e^	349.8 ± 6.2 ^c^	400.5 ± 9.3 ^b^
Stearic acid	16.5 ± 0.5 ^a^	9.2 ± 0.5 ^e,f^	8.6 ± 0.1 ^f^	14.6 ± 0.3 ^b^	1.0 ± 0.1 ^h^	13.2 ± 0.1 ^c^	9.9 ± 0.2 ^d^	7.6 ± 0.2 ^g^	9.3 ± 0.1 ^d,e^
Eicosanoic acid	93.9 ± 1.6 ^b^	84.1 ± 6.0 ^c^	170.9 ± 2.8 ^a^	10.2 ± 0.3 ^d^	5.8 ± 0.3 ^d,e^	8.7 ± 0.2 ^d^	7.2 ± 0.2 ^d^	7.1 ± 0.1 ^d^	0.7 ± 0.0 ^d,e^
Docosanoic acid	43.7 ± 1.0 ^b^	23.6 ± 2.4 ^c^	53.8 ± 1.6 ^a^	41.3 ± 1.1 ^b^	6.7 ± 0.3 ^f^	10.9 ± 0.1 ^e^	16.0 ± 0.1 ^d^	8.2 ± 0.2 ^f^	9.2 ± 0.4 ^e,f^
Tricosanoic acid	27.6 ± 1.1 ^b^	22.7 ± 1.3 ^c^	45.2 ± 0.3 ^a^	20.3 ± 0.3 ^d^	5.5 ± 0.3 ^g^	8.8 ± 0.2 ^e^	7.1 ± 0.8 ^f^	6.6 ± 0.1 ^f,g^	7.7 ± 0.3 ^e,f^
Lignoceric acid	22.1 ± 1.8 ^c^	163.6 ± 5.1 ^b^	618.9 ± 23.1 ^a^	13.9 ± 0.2 ^c^	5.3 ± 0.2 ^c^	11.1 ± 0.5 ^c^	7.2 ± 0.2 ^c^	6.8 ± 0.1 ^c^	7.8 ± 0.4 ^c^
Hyenic acid	32.5 ± 1.0 ^b^	25.8 ± 0.2 ^c^	37.2 ± 1.7 ^a^	8.8 ± 0.2 ^d^	3.3 ± 0.1 ^f^	6.9 ± 0.0 ^e^	6.6 ± 0.1 ^e^	5.6 ± 0.1 ^e^	7.3 ± 0.1 ^d,e^
Cerotic acid	10.0 ± 0.7 ^b^	7.9 ± 0.4 ^c,d^	26.9 ± 1.6 ^a^	10.2 ± 0.3 ^b^	5.8 ± 0.3 ^e^	8.7 ± 0.2 ^b,c^	7.3 ± 0.2 ^c,d^	7.2 ± 0.1 ^d,e^	7.5 ± 0.3 ^c,d^
Montanic acid	34.4 ± 1.3 ^a^	15.2 ± 0.4 ^c^	17.7 ± 0.6 ^b^	9.5 ± 0.2 ^d^	5.4 ± 0.2 ^f^	7.4 ± 0.1 ^e^	9.8 ± 0.2 ^d^	6.1 ± 0.1 ^f^	9.1 ± 0.1 ^d^
Total	838.8 ± 20.6 ^b^	547.0 ± 6.9 ^d^	1341.7 ± 15.0 ^a^	591.6 ± 14.5 ^c^	66.7 ± 2.7 ^i^	119.0 ± 0.5 ^h^	250.4 ± 2.0 ^g^	419.7 ± 7.5 ^f^	476.8 ± 11.0 ^e^
Unsaturated fatty acids									
Oleic acid	610.5 ± 26.0 ^b^	278.4 ± 5.4 ^d^	866.7 ± 20.9 ^a^	339.4 ± 7.6 ^c^	19.9 ± 0.9 ^g^	44.8 ± 0.4 ^g^	213.0 ± 7.1 ^e^	103.6 ± 10.5 ^f^	340.2 ± 6.5 ^c^
Linoleic acid	1079.5 ± 56.5 ^c^	8.8 ± 0.5 ^g^	46.9 ± 0.3 ^g^	749.6 ± 21.4 ^d^	285.0 ± 14.9 ^f^	588.5 ± 16.7 ^e^	593.1 ± 5.1 ^e^	5065.0 ± 93.4 ^b^	5880.9 ± 144.9 ^b^
α-Linolenic acid	94.9 ± 4.4 ^c^	354.6 ± 6.6 ^a^	292.1 ± 1.5 ^b^	24.0 ± 0.3 ^e,f^	11.3 ± 0.5 ^g^	33.2 ± 0.1 ^d^	17.3 ± 2.3 ^f,g^	14.1 ± 0.4 ^g^	25.3 ± 1.1 ^e^
Total	1784.9 ± 78.7 ^c^	641.8 ± 10.3 ^f^	1205.7 ± 22.1 ^d^	1113.1 ± 29.0 ^d^	316.2 ± 15.8 ^g^	666.5 ± 16.4 ^f^	823.4 ± 11.1 ^e^	5183.7 ± 89.5 ^b^	6246.4 ± 147.8 ^a^

Values were presented as normalized area by ribitol or n-tridecane (internal polar and non-polar standards, respectively). CTRL: Control fruits. Different superscript letters (a–g) indicate statistical significance (*p* < 0.05) at the same line (mean ± standard deviation, *n* = 4). GABA, γ-aminobutyric acid.

## Data Availability

https://repositorio.uspdigital.usp.br/ accessed on 15 April 2021.

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
