# Peer review of "Post-Harvest Treatment with Methyl Jasmonate Impacts Lipid Metabolism in Tomato Pericarp (Solanum lycopersicum L. cv. Grape) at Different Ripening Stages"

_foods, 2021, doi:10.3390/foods10040877_

Round 1
Reviewer 1 Report
The manuscript by Meza et al. describes the effects on metabolome profiles (primary and secondary metabolites) when exogenous methyl jasmonate is applied in combination with 1-methylcyclopropane. The article is nicely written, and it reads easily. However, I encourage the authors to work on the following minor issues.
1) Extraction procedures. Lines 94 and 111. The authors discussed the extraction of frozen pericarp. If it is the pericarp what has been studied, then authors should specify how the prepare the sample and how they took the pericarp out of the fruit and how they removed the rest of the tomato fruit. Again, if the authors have just analyzed pericarps, this is an important feature that should be mentioned in Title, Abstract and Conclusions. Authors should specifically mention that their work is focused on pericarp and not in the fruit as a whole.
2) The whole discussion is based on 4, 10 and 21 DAH aiming to observe the effect of treatments with respect to CTRL, BUT, what about inter-treatment differences? If possible, I recommend the authors to include in the analysis 13 DAH, since it is the day in where differences in ethylene emission and fruit color are maximized and then the two treatment’s effect (MCP and MCP+MeJA) could be better explained.
3) Table 1. Error analysis and significant figures. Please revise. The errors should not contain more significant figures than the measurement.
4) Table 1. Descriptors a to h. What are their meanings?
5) Line 238. Replace 21th by 21st.
Author Response
RESPONSE TO REVIEWER #1
March 02nd, 2021
Manuscript ID: foods-1173705
Title: “Postharvest treatment with methyl jasmonate impacts lipid metabolism in tomato pericarp (Solanum lycopersicum L. cv. Grape) at different ripening stages”
Dear Reviewer,
We are pleased to submit our paper after a deep revision attending the recommendations and suggestions of the Foods referees. All of them were taken in account and we are sure that the corrections suggested by the reviewers really contributed to improve the quality of this paper. As recommended by the reviewers, the English grammar has been revised and improved. We also made small corrections which did not alter the context neither the format. Appended to this letter is our point-by-point response in (red) answering the comments raised by each reviewer.
Please activate the “revision mode” in the main text (Word document) to track changes in the revised manuscript (tracked version).
Hopefully, we were able to clarify major criticism and we are looking forward to getting this paper published in the Foods. Also, we would like to thank the reviewers for their valuable revision.
Thank you very much in advance.
Sincerely,
Eduardo Purgatto
Researcher of Food Research Center (FoRC/CEPID/FAPESP)
Department of Food and Experimental Nutrition, Faculty of Pharmaceutical Sciences (FCF), University of São Paulo (USP), São Paulo, SP, Brazil
Reviewer #1:
We thank the reviewer for the corrections and comments. The pages and lines described below correspond to the tracked changes in revised manuscript (tracked version).
Point 1: Lines 94 and 111: Extraction procedures. The authors discussed the extraction of frozen pericarp. If it is the pericarp what has been studied, then authors should specify how the prepare the sample and how they took the pericarp out of the fruit and how they removed the rest of the tomato fruit. Again, if the authors have just analyzed pericarps, this is an important feature that should be mentioned in Title, Abstract and Conclusions. Authors should specifically mention that their work is focused on pericarp and not in the fruit as a whole.
Response: Authors: Accepted. We thank the reviewer for noticing this need for reformulation related to the extraction procedure.
The preparation details of the sample were added as recommended by the reviewer
Page 02, line 76: Biological replicates were composed of pericarp tissues by removing the placenta and seeds of the fruit. The pericarp samples were frozen in liquid nitrogen and stored at -80 ºC for subsequent analyses.
The information was mentioned in the Title as recommended by the reviewer:
Page 01, line 03: The title was reformulated as suggested: “Postharvest treatment with methyl jasmonate impacts lipid metabolism in tomato pericarp (Solanum lycopersicum L. cv. Grape) at different ripening stages”.
The information was mentioned in the Abstract as recommended by the reviewer:
Page 01, line 21: Pericarp of the fruits were analyzed at 4, 10 and 21 day after harvest (DAH) and compared with the no-treated fruits.
The information was mentioned in the section 3.4. Lipid metabolism affected by the postharvest jasmonate treatment as recommended by the reviewer:
Page 17, line 404: The metabolite profiling of the pericarp of tomato fruits treated with only 1-methylciclopropene and with both 1-methylciclopropene and methyl jasmonate showed a significant impact to the fruit quality and, consequently, to the ripening process.
The information “tomato pericarp” was mentioned in the legends of the Figures 2, 3, 4 ,5 and 6; Table 1, and supplementary file (Table S1 and S2).
Point 2: The whole discussion is based on 4, 10 and 21 DAH aiming to observe the effect of treatments with respect to CTRL, BUT, what about inter-treatment differences? If possible, I recommend the authors to include in the analysis 13 DAH, since it is the day in where differences in ethylene emission and fruit color are maximized and then the two treatment’s effect (MCP and MCP+MeJA) could be better explained.
Response: We agree that the inclusion of analyzes for the 13th day after harvest would be interesting and we are grateful for the reviewer's suggestion. In our experimental design, we chose to follow criteria like those used in other studies with tomato fruit as a model. We set the control group as our parameter and chose to collect the samples in the Mature Green, Breaker and Red ripe stages. For the treated groups, we collected the fruits at points equivalent to the same color changes. We understand that in this design, intermediate points would no longer be verified, but we made this decision to balance the volume of data generated by the analysis of the metabolic profile plus the representativeness necessary for a sample of fruits harvested per group (100 fruit each, 3 groups) multiplied by the number of biological replicates (4), which corresponds to four independent experiments. Although the fruits were grown in a greenhouse under controlled conditions, we chose to carry out the experiments with a short temporal difference to minimize possible seasonal effects. For the ripening monitoring, we carried out measurements of ethylene and color on other days than those established by the harvest criteria, but we followed the original sampling plan. Unfortunately, we do not have samples taken on the 13th. We fully agree with the reviewer's suggestion, and we understand that represent more information to the set of data we have obtained, and an interesting addition to the discussion. However, face of the lack of the sample, we would like to request the understanding of the reviewer and the editor and consider the experimental design and the results as they are presented in the article.
Point 3: Table 1. Error analysis and significant figures. Please revise. The errors should not contain more significant figures than the measurement.
Response: Accepted. Page 07, line 237: Table1. Measurement and error analysis were adjusted with one significant figure.
Point 4: Table 1. Descriptors a to h. What are their meanings?
Response: Accepted. Page 07, line 237: Table1. Descriptors A to E have been added to Table 1 to provide a specific order in the table and in the text. However, they were removed o make the information clearer. In the revised version of our paper, the different superscript letters at Table 1 indicate statistical significance.
Point 5: Line 238. Replace 21th by 21st.
Response: Accepted. Page 11, line 251: The word “21th” was substituted by “21st” as recommended.
Point 6: Small corrections in the section 3.3 (Secondary metabolite profiling affected by postharvest hormonal treatment) were made in order to improve the reading. However, any of these modifications did not alter the context of the main text.
Response: Page 15, line 334: Lycopene was the most affected by the action of 1-methylciclopropene, reducing its level not only in MCP, but also in MCP+MeJA by 29 and 25-fold, respectively, at 4 DAH, while at 10 DAH lycopene was reduced by 8 and 6-fold, respectively, when compared with CTRL (Figure 6A). β-carotene and lutein showed an decrease lesser than 3-fold by 1-methylciclopropene at ripening stages (Table S1) These remarkable impact on the synthesis of carotenoids can be observed in the Figure 2, mainly at the onset of ripening.
Page 15, line 353: However, the action of 1-methylciclopropene had a lesser impact on lycopene accumulation at 21 days of hormonal treatment by decreasing 2.8-fold its production, and its action was totally reversed by the methyl jasmonate hormone. Fruits treated with methyl jasmonate showed an increase not only in lycopene production, but also in β-carotene and lutein at 21 DAH, indicating the important role that methyl jasmonate play in the synthesis of carotenoids (Figure 6A). Lycopene and β-carotene showed an increase of 10 %, and lutein of 20% when compared with CTRL (Figure 6A, Table S1), which is considered relevant once these bioactive compounds have been associated with health benefits leading to decreases in the occurrence of chronic non-communicable diseases [25]. The total carotenoids level was represented mainly by lycopene.
Reviewer 2 Report
The authors should better describe the different factors describing quality of products and mention related references such as:
Durazzo et al. 2010. Influence of different crop management practices on the nutritional properties and benefits of tomato -Lycopersicon esculentum cv Perfectpeel- International Journal of Food Science and Technology 45, 2637–2644. ISSN: 0950-5423, DOI:10.1111/j.1365-2621.2010.02439.x.
Data in Figure 1 should be better described in the text
Check expression of values in Table 1
The description of 3.2. Primary metabolite profiling affected by postharvest hormonal treatment should be implemented.
A section conclusion including limits, advantages, practical applications and future directions should be inserted.
Author Response
RESPONSE TO REVIEWER #2
March 02nd, 2021
Manuscript ID: foods-1173705
Title: “Postharvest treatment with methyl jasmonate impacts lipid metabolism in tomato pericarp (Solanum lycopersicum L. cv. Grape) at different ripening stages”
Dear Reviewer,
We are pleased to submit our paper after a deep revision attending the recommendations and suggestions of the Foods referees. All of them were taken in account and we are sure that the corrections suggested by the reviewers really contributed to improve the quality of this paper. As recommended by the reviewers, the English grammar has been revised and improved. We also made small corrections which did not alter the context neither the format. Appended to this letter is our point-by-point response in (red) answering the comments raised by each reviewer.
Please activate the “revision mode” in the main text (Word document) to track changes in the revised manuscript (tracked version).
Hopefully, we were able to clarify major criticism and we are looking forward to getting this paper published in the Foods. Also, we would like to thank the reviewers for their valuable revision.
Thank you very much in advance.
Sincerely,
Eduardo Purgatto
Researcher of Food Research Center (FoRC/CEPID/FAPESP)
Department of Food and Experimental Nutrition, Faculty of Pharmaceutical Sciences (FCF), University of São Paulo (USP), São Paulo, SP, Brazil
Reviewer #2:
We thank the reviewer for the corrections and comments. The pages and lines described below correspond to the tracked changes in revised manuscript (tracked version).
Point 1: The authors should better describe the different factors describing quality of products and mention related references such as: Durazzo et al. 2010. Influence of different crop management practices on the nutritional properties and benefits of tomato -Lycopersicon esculentum cv Perfectpeel- International Journal of Food Science and Technology 45, 2637–2644. ISSN: 0950-5423, DOI:10.1111/j.1365-2621.2010.02439.x.
Response: Authors: Accepted. We thank the reviewer for noticing this need of information.
Page 17, line 426: Moreover, notable accumulations in the levels of secondary metabolites, such as lycopene, β-carotene, lutein, α-, β- and γ-tocopherols, β-sitosterol, estigmasterol and estigmastadienol were detected at 21 DAH by the action of methyl jasmonate (Figure 6, Table S1 and S2). However, it is important to note that the maturation stage and hormonal regulation may not be the only factors responsible for the improvement in lipid metabolism in tomato fruits, and other factors such as genetic factors, cultural practices, cultivation and environmental conditions should be considered [25]. At that time, understanding the interaction among hormonal treatment and environmental factor, genotype and agronomic practices is essential to produce high quality fruits by the improvement in the synthesis of high-valued nutrients.
Page 20, line 541: Reference was added at the reference list. 25. Durazzo, A. Azzini, E. Foddai, M.S. et al. (2010). Influence of different crop management practices on the nutritional properties and benefits of tomato -Lycopersicon esculentum cv Perfectpeel-. International Journal of Food Science and Tech-nology, 45, 2637-2644. doi:10.1111/j.1365-2621.2010.02439.x.
Point 2: Data in Figure 1 should be better described in the text
Response: Authors: Accepted. We are grateful for your constructive criticism. The discussion of the data was improved as described below:
Page 06, line 218: Primary metabolites are important components related to the fruit quality. In addition, they are are considered crucial for plant growth and development. For this fact, understanding the metabolism of the fruit can support the finding of future approaches for its manipulation [24]. In this work, a total of 46 primary metabolites were identified by metabolomics analysis: 10 sugars (glucose, fructose, sucrose, allose, gulose, glucaric acid, myo-inositol, mannose, ribose, and arabinofuranse); 9 organic acids (oxaloacetic, citric, succinic, aconitic, malic, citraconic, fumaric, propanoic, and butanoic acids); 12 amino acids (proline, serine, valine, threonine, aspartic acid, glutamic acid, glutamine, γ-aminobutyric acid (GABA), asparagine, tryptophan, phenylalanine, and tyrosine); 12 saturated fatty acids (capric, lauric, myristic, palmitic, stearic, eicosanoic, docosanoic, tricosanoic, lignoceric, hyenic, cerotic, and montanic acids); and 3 unsaturated fatty acids (oleic, linoleic, and linolenic acids) at 4, 10 and 21 DAH (Table 1). In Table 1, the effects of methyl jasmonate and 1-methylcyclopropene on the accumulation or reduction of each metabolite in the three different stages of maturation can be seen from the normalized area by the internal standard.
Point 3: Check expression of values in Table 1
Response: Authors: Accepted. Table 1 was checked and corrected as recommended by the reviewer. Measurement and error analysis were also adjusted with one significant figure. (Page 07, line 237).
Point 4: The description of 3.2. Primary metabolite profiling affected by postharvest hormonal treatment should be implemented.
Response: Authors: Accepted. The section 3.2. Primary metabolite profiling affected by postharvest hormonal treatment was implemented as recommended by the reviewer.
Page 13, line 278: Primary metabolism is essential for the fruit quality. Sugars, organic acids and amino acids are responsible for the taste of tomato fruits, providing the sensorial perception. Amino acids and fatty acids play an important role as precursors of aroma compounds [7]. Treatment with 1-methylcyclopropene impacted sugar and organic acids, inhibiting their production during ripening. Fruits treated only with 1-methylcyclopropene were most affected, showing a greater delay in accumulate sugars and organic acids than those fruits treated with both 1-methylcyclopropene and methyl jasmonate (Figure 3). For in-stance, glucose showed a significantly reduction of 22, 13 and 23-fold at 4, 10 and 21 DAH, respectively, in MCP when compared with CTRL. Mannose, ribose, malic and aconitic acids exhibited a decrease in their levels of 14, 30, 21 and 20-fold at 4 DAH, whereas fructose, sucrose and citraconic acid showed 12, 15 and 27-fold lower levels at 10 DAH when compared with CTRL (Table 1). Reductions in the levels of these metabolites in treated fruits with 1-methylciclopropene are clearly observed in the Figure 2 by the fold change analysis between treated fruits and control group.
Page 14, line 294: Exceptionally, levels of glucose, glucaric acid and mannose showed an increase at 10 DAH in MCP+MeJA when compared to CTRL. Similar behavior was observed by myo-inositol, propanoic and butanoic acids at 21 DAH (Table 1,). In the Figure 3, heatmap analysis showed a clearly tendency of these metabolites to increase at 10 DAH. As observed by ethylene emission, the minor impact on the production sugars and organic acids observed by MCP+MeJA may suggest that methyl jasmonate play an important role in ripening process, which may act independently of endogenous eth-ylene, or a stimulation of the synthesis of new receptors, or the blockage of ethylene receptors were reversed after some period.
Page 14, line 303: Amino acids profiling were also affected by the action of 1-methylcyclopropene. An inhibition in the production of amino acids during ripening were observed in both MCP and MCP+MeJA when compared with control (Figure 4). The most affected amino acids were aspartic acid at 4 DAH and GABA at 10 DAH, showing a reduction in their levels of 28 and 10 fold in MCP, respectively, while MCP+MeJA showed 11 and 14 fold decreased, respectively, when compared with CTRL, which clearly observed in the Figure 2. In contrast, tyrosine and phenylalanine showed levels 2 and 9-fold higher in MCP and MCP+MeJA at 4 DAH when compared with CTRL (Table 1, Figure 2). It is important to highlight that phenylalanine and tyrosine are aromatic amino acids, which participate of shikimate pathway and are responsible for aroma development of fruit. In the Table 1 is showed that the total amino acids level was represented mostly by proline, glutamic and aspartic acids, which are important to fruit quality by providing the sweetness and umami taste.
Page 14, line 316: In addition, fatty acids profiling was also affected by the postharvest treatments. The action of 1-methylciclopropene showed a greater impact on fatty acids such as oleic, capric, lauric, palmitic, stearic and myristic acids at 10 DAH as showed in the Figure 5, decreasing their levels 17, 10, 14, 17, 14 and 12fold in MCP group, respectively, and 7, 6, 9, 11, 1, and 7-fold in MCP+MeJA, respectively, when compared with CTRL (Table 1). The reduction in fatty acids by 1-methylciclopropene is evidently when the fold change analysis was applicable (Figure 2). MCP+MeJA group also showed a reduction in fatty acids levels, but they were lesser impacted when compared with MCP group (Figure 4). However, the most impacted was the linoleic and myristic acids at 4 DAH with a reduction of 119 and 26-fold in MCP, respectively, and 23 and 9 in MCP+MeJA, respectively, when compared with CTRL (Table 1).
Page 14, line 336: Palmitic and eicosanoic acids contributed essentially with the total of saturated fatty acids level, whereas oleic and linoleic acids with the total of unsaturated fatty acids level, which is notable as they play an important role for the fruit quality and nutritional value.
Point 5: A section conclusion including limits, advantages, practical applications and future directions should be inserted.
Response: Authors: Accepted. Page 18, line 446: The treatment with methyl jasmonate can induce significant changes in the metabolite profile in tomato fruits during ripening, with positive impact on the nutritional and sensorial quality of the fruit. This treatment, associated with the blocking of ethylene receptors with 1-methylcyclopropene, proved to be effective in avoiding potential effects on the postharvest life of the fruits, due to the increase in ethylene synthesis caused by methyl jasmonate. Our experimental design involved four experiments, and the consistency obtained in the results indicates that the effects have good reproducibility. However, they were carried out only in the Grape variety, and it would be interesting to reproduce these treatments in other cultivars to assess the influence of the genotype on responses to treatment with methyl jasmonate and 1-methylcyclopropene. From the point of view of applicability, the presented protocol has a good commercial potential, since the concentrations of methyl jasmonate and 1-methylcyclopropene used were low and, consequently, did not use large volumes of the compounds. The volatility of the compounds plus the simplicity of the method of exposure of the fruits make the treatments feasible for larger environments, such as commercial chambers. Although both substances are considered GRAS (generally recognized as safe), further studies about the impact on sensory quality would be important to assess consumer acceptability for treated fruit.
Point 6: Small corrections in the section 3.3 (Secondary metabolite profiling affected by postharvest hormonal treatment) were made in order to improve the reading. However, any of these modifications did not alter the context of the main text.
Response: Page 15, line 334: Lycopene was the most affected by the action of 1-methylciclopropene, reducing its level not only in MCP, but also in MCP+MeJA by 29 and 25-fold, respectively, at 4 DAH, while at 10 DAH lycopene was reduced by 8 and 6-fold, respectively, when compared with CTRL (Figure 6A). β-carotene and lutein showed an decrease lesser than 3-fold by 1-methylciclopropene at ripening stages (Table S1) These remarkable impact on the synthesis of carotenoids can be observed in the Figure 2, mainly at the onset of ripening.
Page 15, line 353: However, the action of 1-methylciclopropene had a lesser impact on lycopene accumulation at 21 days of hormonal treatment by decreasing 2.8-fold its production, and its action was totally reversed by the methyl jasmonate hormone. Fruits treated with methyl jasmonate showed an increase not only in lycopene production, but also in β-carotene and lutein at 21 DAH, indicating the important role that methyl jasmonate play in the synthesis of carotenoids (Figure 6A). Lycopene and β-carotene showed an increase of 10 %, and lutein of 20% when compared with CTRL (Figure 6A, Table S1), which is considered relevant once these bioactive compounds have been associated with health benefits leading to decreases in the occurrence of chronic non-communicable diseases [25]. The total carotenoids level was represented mainly by lycopene.